# Prediction of White Matter Hyperintensity in Brain MRI Using Fundus Photographs via Deep Learning

**DOI:** 10.3390/jcm11123309

**Published:** 2022-06-09

**Authors:** Bum-Joo Cho, Minwoo Lee, Jiyong Han, Soonil Kwon, Mi Sun Oh, Kyung-Ho Yu, Byung-Chul Lee, Ju Han Kim, Chulho Kim

**Affiliations:** 1Department of Ophthalmology, Hallym University Sacred Heart Hospital, Anyang 14068, Korea; bjcho8@gmail.com (B.-J.C.); magicham@naver.com (S.K.); 2Medical Artificial Intelligence Center, Hallym University Medical Center, Anyang 14068, Korea; jiyong0719@mach.hallym.or.kr; 3Division of Biomedical Informatics, Seoul National University Biomedical Informatics (SNUBI), Seoul National University College of Medicine, Seoul 03080, Korea; 4Department of Neurology, Hallym Neurological Institute, Hallym University Sacred Heart Hospital, Anyang 14068, Korea; minwoo.lee.md@gmail.com (M.L.); iyyar0408@gmail.com (M.S.O.); ykh1030@gmail.com (K.-H.Y.); ssbrain@hallym.or.kr (B.-C.L.); 5Department of Neurology, Chuncheon Sacred Heart Hospital, Chuncheon 24253, Korea

**Keywords:** cerebral small-vessel disease, fundus photograph, white matter hyperintensity, Fazekas scale, deep learning

## Abstract

Purpose: We investigated whether a deep learning algorithm applied to retinal fundoscopic images could predict cerebral white matter hyperintensity (WMH), as represented by a modified Fazekas scale (FS), on brain magnetic resonance imaging (MRI). Methods: Participants who had undergone brain MRI and health-screening fundus photography at Hallym University Sacred Heart Hospital between 2010 and 2020 were consecutively included. The subjects were divided based on the presence of WMH, then classified into three groups according to the FS grade (0 vs. 1 vs. 2+) using age matching. Two pre-trained convolutional neural networks were fine-tuned and evaluated for prediction performance using 10-fold cross-validation. Results: A total of 3726 fundus photographs from 1892 subjects were included, of which 905 fundus photographs from 462 subjects were included in the age-matched balanced dataset. In predicting the presence of WMH, the mean area under the receiver operating characteristic curve was 0.736 ± 0.030 for DenseNet-201 and 0.724 ± 0.026 for EfficientNet-B7. For the prediction of FS grade, the mean accuracies reached 41.4 ± 5.7% with DenseNet-201 and 39.6 ± 5.6% with EfficientNet-B7. The deep learning models focused on the macula and retinal vasculature to detect an FS of 2+. Conclusions: Cerebral WMH might be partially predicted by non-invasive fundus photography via deep learning, which may suggest an eye–brain association.

## 1. Introduction

Cerebral small-vessel disease (SVD), including white matter hyperintensity (WMH), is associated with an increased risk of stroke, cognitive decline, cardiovascular disease (CVD), and mortality [1,2]. This change is related to aging and hypertension, and is frequently observed on the brain magnetic resonance imaging (MRI) of elderly people [3]. Interestingly, cerebral small vessels share physiological characteristics, such as size and ontogeny, with retinal vessels, which are part of the brain’s vascular system [4,5]. The retinal vasculature has been reported to reflect the concurrent neurovascular status and cerebral SVD burden in the brain [6,7,8]. Fortunately, retinal vascular features, which include vessel caliber, fractal dimensions, and tortuosity, can easily be examined by fundus photography of the eye in a non-invasive way [9]. Fundus photographs can directly capture the retinal vessels and the optic nerve, even in a non-mydriatic state, within a short time. The identification of pathological retinal structures in fundus photographs may help predict the pathological status of the brain.

Furthermore, with recent advances in computing power and storage devices, deep learning (DL) algorithms, an application of artificial intelligence, have been widely adopted in medical research, including in medical image evaluation, sentiment analysis using medical texts, and outcome prediction using sequential signal data [10,11]. As DL algorithms may accurately incorporate multiple imaging features that are invisible to humans [12,13], previous studies have utilized several DL models to reveal the predictability of fundus photography on a variety of ocular diseases, including age-related macular degeneration, glaucoma, and central retinal vein occlusion [14,15]. Recently, retinal characteristics found using machine learning models for automatic retinal image analysis (ARIA) have been found to predict WMH and its classification in healthy adults [16]. However, no study has yet investigated whether raw fundoscopy images that use a DL approach have potential for predicting the degree of cerebral WMH in the general population.

Therefore, in this study, we investigated whether DL applied to fundus photographs can predict cerebral SVD burden, which is represented as WMH on brain MRI. Cerebral SVD was evaluated using a modified Fazekas scale (FS) that has been used as a grading system for cerebral WMH in several clinical trials [17,18]. In addition, we investigated which features in the fundus photographs were associated with the degree of WMH. To meet this need, a salience map that visualized the region of interest for a DL model was implemented and analyzed by an experienced retina specialist. To the best of our knowledge, this is the first study to use convolutional neural networks to examine the relationship between retinal fundoscopic images and cerebral WMH.

## 2. Materials and Methods

### 2.1. Study Participants

This retrospective, exploratory study was conducted in a tertiary referral center. The flowchart of participant enrollment is presented in Figure 1. We screened patients who had undergone fundus photography as part of regular health examinations at Hallym University Sacred Heart Hospital between January 2010 and June 2020. Among the patients, those who had undergone brain MRI, either as a regular screening in the health promotion center or for any reason at the institution, were selected. Only subjects who also had brain MRI results available within one year (before or after) of the fundus photography were included. Patients who had no fluid-attenuated inversion recovery (FLAIR) sequence on their brain MRI and those who had fundoscopic images of poor quality, such as images that were defocused, blurred, opaque, or dark enough to obscure retinal vessels, were excluded. Since this study’s aim was not to assess retinopathic changes in patients with confirmed SVD on MRI, there is a time difference between the MRI and fundoscopic examination. As the time of fundoscopy is later than the time of MRI, fundoscopic changes may be overestimated in the ML model using an MRI dataset. Therefore, we only included subjects with a time gap between MRI and fundus imaging within 1 year [19].

This study was performed in accordance with the Declaration of Helsinki, and it was approved by the Institutional Review Board, which waived the requirement for written informed consent (IRB No. 2018-12-031).

### 2.2. Grading Cerebral White Matter Hyperintensities

To sensitively differentiate WMH from other lesions [20], the FLAIR sequence of the brain MRI was assessed using two different scanners (Skyra 3.0T, Siemens, Erlangen, Germany and Achieva 3.0T, Philips, Best, The Netherlands). All brain MRI scans were performed using a standard head coil with the following parameters: TR, 9000–11,000 range; TE 120–130 range; matrix size, 256 × 256 pixels; field of view, 230 × 230 mm; slice thickness, 5 mm; inter-slice gap, 1 mm.

The degree of cerebral WMH on the FLAIR MRI was determined using modified Fazekas criteria that ranged from 0 to 3. The detailed criteria for the FS are described elsewhere [21]. The degree of cerebral WMH was determined by two vascular neurologists (M.L. and C.K.), and any discordant results between the two observers were resolved by consensus (Appendix A). For each patient, only one brain MRI scan—with the highest FS value and shortest period to the closest fundus photography—was selected.

### 2.3. Collection of Fundus Photographs 

Fundus photographs were taken at the health promotion center without pupillary dilation in a dark room using a nonmydriatic color fundus digital camera (Kowa Nonmyd 7, Kowa Company Ltd., Tokyo, Japan). The macula was centered in the fundus photograph with a field of view of 45°, and the photograph was acquired at a resolution of 2592 × 3872 pixels. To prevent the effect of non-lesioned personal retinal characteristics, only one photograph was taken for each eye of each participant on one examination date. Then, for each eye of each patient, only one fundus photograph taken at the closest point in time to the selected MRI scan was included, which ensured that the time between the fundus photography and the MRI scan was less than 1 year. Demographic metadata, including age and sex at the time of the fundus image capture, were also collected to adjust for age-related changes in the fundus photographs. 

### 2.4. Construction of Datasets

To train the DL models, all the included fundus photographs were matched to the results of the FS assessed on the corresponding brain MRI scan. Because the FS grades of the patients were severely unbalanced in size (FS grade 0: 1: 2: 3 = 1019: 641: 189: 43), FS grades 2 and 3 were merged into one group and labeled “FS grade 2+”.

The main outcome measure was predictive performance for the presence of WMH. WMH was regarded to be present at FS grade 1 or higher only, as defined by the Fazekas scale [21]. The secondary outcome measure was the predictive performance for FS grades 0, 1, or 2+ based on fundus photographs. Notably, there was a significant between-group data imbalance for patient age and data size (Appendix A). Therefore, for the FS grade prediction, to verify predictability and avoid age bias, FS grade 0, 1, and 2+ groups were age-matched via random undersampling (Figure 1). For this dataset balancing, all patients were divided into age subgroups at intervals of 5 years, which started from the 15–19 years age subgroup and ended at the 85–89 years age subgroup. Then, for each age subgroup, random sampling was performed the same number of times for each FS grade class, by the patient number of the smallest class.

To examine the model’s performance robustly, a 10-fold cross-validation method was adopted. The entire dataset was randomly divided into ten groups using patients’ identity as the selection key. A random split was performed for each FS grade class to ensure that the divided groups represented a consistent proportion of classes. For each fold, one-ninth of the training dataset was allocated to the validation dataset (or tuning dataset), stratified by class, which was not used in the training process, but was used in the parameter-tuning process.

### 2.5. Data Pre-Processing & Augmentation

All images were centrally cropped to remove bilateral black empty spaces and then resized to 740 × 648 pixels. Contrast-limited adaptive histogram equalization was applied to all images [22]. The images were normalized with mean and standard deviation values from the ImageNet database. Data quantity augmentation was not performed because data doubling by horizontal flipping did not increase model performance in our pilot studies.

### 2.6. Training Convolutional Neural Networks

For transfer learning, the DenseNet-201 and EfficientNet-B7 architectures, which were pre-trained using the ImageNet dataset, were adopted [23,24]. Briefly, DenseNet uses a dense block structure that concatenates the feature maps of previous layers [23], and EfficientNet-B7 uses MBConv blocks balanced for width and depth by reinforcement learning as a state-of-the-art algorithm [24]. The architectures open to the public were adopted: DenseNet-201 (https://pytorch.org/hub/pytorch_vision_densenet; accessed on 12 October 2021) and EfficientNet-B7 (https://github.com/lukemelas/EfficientNet-PyTorch; accessed on 12 October 2021).

When we trained this model, we used the Adam optimizer with a β1 of 0.9 and a β2 of 0.999. For hyperparameter tuning, genuine grid search strategies were not applied. However, we explored the best-performing condition of hyperparameters by sequential reduction. For batch size, the largest size that the memory of the graphics processing unit (GPU) could deal with for EfficientNet-B7 in our server was 6. We explored the performance of each batch size, decreasing it to 6, 4, and 3, in our pilot studies, and then adopted the best-performing batch size, 4. After fixing the batch size, the learning rate was explored, starting from 1e^−4^ and then being sequentially reduced by 0.1. Finally, the learning rate 1e^−6^, which showed the highest performance, was adopted. The initial learning rate was 1e^−6^, which then decayed by 0.1 every 10 epochs until the learning rate reached 1e^−10^. Categorical or binary cross-entropy loss for the best accuracy was used as the loss function. Cross-entropy function is widely used in deep neural network classification, in which cross-entropy effects are minimized to discriminate between correct and competing classes in a non-linear fashion. [25]. The batch size was set to 4. Early stopping was used with a patience of 30 epochs after the first 40 epochs. The training was performed on the PyTorch platform using NVIDIA Titan RTX GPUs, Intel i9-9920X central processing units, and a 128 GB RAM server.

### 2.7. Implementation of Salience Maps

To visualize the region of interest of the DL models to predict FS grade, gradient-weighted class activation mapping (Grad-CAM) was adopted. This method provides an inference-level visualization technique for image classification, with which we can infer which image characteristics are important for image DL classification [26]. An attention map was implemented to present the region of interest. To realize the map, the last two layers of the classifier were removed, and then a global average pooling layer and sigmoid layer were added instead. All attention maps for the test dataset were reviewed by an experienced retinal specialist (B.-J.C.).

### 2.8. Statistical Analysis

Continuous variables are expressed as means ± standard deviation, and categorical variables are presented as numbers and percentages. Inter-rater agreement between the two observers for FS labeling was measured using intraclass correlation coefficients. The performance of DL classifiers was evaluated using the area under the receiver operating characteristic curve (AUC). Sensitivity, specificity, positive predictive value (PPV), and negative predictive value (NPV) were calculated at the point on the receiver operating characteristic (ROC) curve that maximized Youden’s J statistic [27]. Statistical analysis was performed using the Python scikit-learn package version 0.18.2 (http://www.python.org; accessed on 5 October 2021).

## 3. Results

A total of 3726 fundus photographs from 1892 patients were included in the study. The numbers of patients with FS grades 0, 1, and 2+ were 1019, 641, and 232, respectively. The data composition is presented in Table 1. The mean age and proportion of male subjects in the whole dataset were 62.4 ± 6.9 and 54.5%, respectively. The inter-rater reliability for FS grades 0, 1 or 2, and 3 was excellent (intraclass correlation coefficient: 0.919, *p* < 0.001, Appendix A). For the FS grade prediction, 905 fundus photos from 462 age-matched patients were included.

### 3.1. Prediction of the Presence of WMH

The mean AUC for predicting WMH presence was 0.736 ± 0.030 on DenseNet-201 and 0.724 ± 0.026 on EfficientNet-B7. The ROC curves of the best-performing folds are shown in Figure 2, and the confusion matrices for the best-performing folds are shown in Figure 3. The mean accuracies for the binary classification were 67.6 ± 2.1% on DenseNet-201 and 67.4 ± 2.4% on EfficientNet-B7. The diagnostic model performances are presented in Table 2. The diagnostic precision values for each fold are presented in Table 3, and the confusion matrices of the best-performing folds are shown in Figure 3. Additional results of the performance of each cross-validation fold for 10- and 5-fold cross-validation of this binary classification are presented in Appendix A.

### 3.2. Falsely Classified Cases

DenseNet-201 and EfficientNet-B7, which predicted FS grades with fundus photograph input, overestimated the FS grades for the test dataset (Figure 2). In the case of the DenseNet-201 algorithm, there was no difference in risk factor profiles related to WMH, such as hypertension and diabetes. However, the age distribution of each cell in the confusion matrix was quite different (Appendix A).

### 3.3. Prediction of Fazekas Scale Grade

The mean accuracy of the three-class prediction of FS, which graded the score as 0, 1, or 2+, reached 41.4 ± 5.7% on DenseNet-201 and 39.6 ± 5.6% on EfficientNet-B7. The reference model, which used the majority voting strategy, achieved accuracy of only 33.5 ± 0.4%. The best-performing folds achieved accuracies of 46.7% and 48.9% on DenseNet-201 and EfficientNet-B7, respectively, and the confusion matrices are presented in Figure 4. Additional results of the performance of each cross-validation fold in 10- and 5-fold cross-validation of the three-class classification are presented in Appendix A.

### 3.4. Salience Maps for High-Grade FS

Representative examples of Grad-CAM are shown in Figure 5. When FS grade 2+ was detected, the CNN was based on the macula and retinal vasculature. Middle- or small-sized vessels, rather than large vessels, around the optic disc are highlighted in Figure 5.

## 4. Discussion

In this study, retinal fundus photographs were shown to partly help predict the degree of WMH in the brain, even in the absence of patient information, including cardiovascular risk factors. The DL model improved the three-class prediction performance by approximately 8% compared with majority voting and was able to differentiate abnormal FS values on brain MRI (mean AUC 0.733) without any information relating to patient characteristics. Although its performance was not remarkable, this result could be interpreted as an indication that the state of the brain can be partly predicted by retinal fundus photography.

Previous studies have also revealed significant associations between the characteristics of retinal vasculature and cerebral SVD [28,29,30]. Retinal vasculature has also been shown to potentially differentiate distinct subtypes of SVD [31,32]. However, previous studies have generally focused on measurable retinal vascular signs or morphology, such as fractal dimension, the tortuosity and branching coefficients of arterioles and venules, and retinal vessel measurements. One study that incorporated a machine learning approach using multiple retinal characteristics with ARIA first analyzed the images based on multiple measurable variables [16]. Thus, our machine learning approach using raw fundoscopic images intended to find novel associations between fundoscopic findings and SVD by analyzing the results of a salience map. Our CAM results were consistent with those of previous studies that investigated the association between fundoscopy and CVD risk factors. They revealed that the DL models used anatomical characteristics, including those of the optic disc and blood vessels, for the prediction of high baseline systolic blood pressure and major adverse cardiovascular events [33].

In our data, the CNN-based DL architecture, especially DenseNet-201, successfully predicted the degree of arteriosclerosis in the brain. The mechanism by which the DL model predicted this burden is unclear. However, this DL algorithm can learn local fundoscopic changes, such as microvascular changes, within a small convolutional layer. DenseNet-201 mainly comprises repetitive convolution, dense blocks, and maximum pooling layers within its main architecture [23]. Unlike the LeNet5 and ResNet (residual neural network) algorithms, a feature map of all leading layers is used as an input for each layer, and each feature map is used as an input for all the subsequent layers. This architecture has been reported to alleviate the vanishing-gradient problem, promote robust feature deployment, encourage feature use, and significantly reduce the number of parameters, which eventually improves the algorithmic performance [34].

Our CNN algorithms showed slightly lower sensitivity for predicting high FS values in two- and three-class classification. In other words, the predicted modified FS values were lower than the actual modified FS values. The performance of our algorithm could be improved, but the current results indicate the following possibilities if we consider the results of previous studies on SVD. Previous pathology studies on WMH have revealed that WMH development is not solely linked to demyelination or axonal loss due to ischemic change [35]. Subtler WMH has also been associated with endothelial and microglial activation, implicating the association of WMH with neuroinflammation [36]. These complex pathomechanisms of WMH on brain MRI may have reduced the power and sensitivity of our DL models using fundoscopy. In further studies on SVD and retinopathy, retinal vessel changes were evident only after overt SVD development, such as cognitive impairment. Nunley et al. investigated 119 patients over 18 years of age to analyze the long-term relationship between retinal vessel changes and cognitive impairment in patients with type 1 diabetes [37]. They found that changes in retinal vessels were observed earlier in patients with cognitive decline related to SVD than in patients with normal cognition. That is, retinal vessel change was more closely related to actual CVD disease development than a surrogate CVD marker, such as SVD. The salience map also changed as a result of two-class classification. On the salience map, the macula and blood vessels were identified as regions of interest playing an important role in WMH classification. In other words, binary classification of WMH would be more affected by changes in the macular or blood vessels than pathological change in the entire retina. Recently, it has been reported that the risk of CVD can be predicted more sensitively by DL when precise ocular images are used, such as optical coherence tomography, rather than fundus images [38]. We assume that DL prediction of SVD burden using additional ophthalmic images will improve our model’s performance in later trials.

The present study has several limitations. First, we did not consider the potential confounding effects of clinical variables, including vascular risk factors, that might have affected both retinal vasculature and cerebral small-vessel changes. However, as we only recruited neurologically healthy patients and excluded those with apparent signs of silent cerebral infarction on brain MRI, our findings may not preclude generalizability. Second, we enrolled subjects whose brain MRI had been performed within 1 year of retinal fundoscopy. Temporal changes in cerebral SVD may have affected the prediction performance, because there is a time interval between brain MRI and ophthalmologic examinations. Despite these limitations, our study has several strengths. First, other studies have reported the usefulness of image DL algorithms in which only normal fundus and definite abnormal fundus images were included (central retinal vein occlusion, diabetic retinopathy, age-related macular degeneration, etc.) [39,40,41]. However, we included all cases of fundus images to train the DL model. It would be more practical to investigate the associations between imaging biomarkers and certain extraocular diseases. Second, we investigated the association between the fundus and other central nervous system disorders. Broadly speaking, the eye, or fundus, is a window through which we can look into the brain directly with the naked eye. In the future, studies on larger samples recruited from outpatient clinics would be required to attain a larger number of patients with a high FS grade (2 or 3). In addition, various diseases showing detectable changes on brain MRI, such as Alzheimer’s disease, could be investigated for associations with retinal changes using DL. There will be more reports on the relevance of fundus image DL and its predictive value for other extraocular or systemic diseases in future studies. Our research methods may serve as a framework for basic analysis methods in such studies.

## 5. Conclusions

In the present study, we demonstrated that raw fundoscopic images can be used to predict cerebral SVD using DL approaches that incorporate convoluted neural network methods. Among the multiple models, DensNet-201 had the highest predictive power for SVD estimation. Both models predicted SVD by observing the optic disc and retinal vessels according to the Grad-CAM results. Future prospective studies that use clinical and demographic variables along with retinal fundoscopic imaging may improve the predictive power of cerebral SVD.

## Figures and Tables

**Figure 1 jcm-11-03309-f001:**
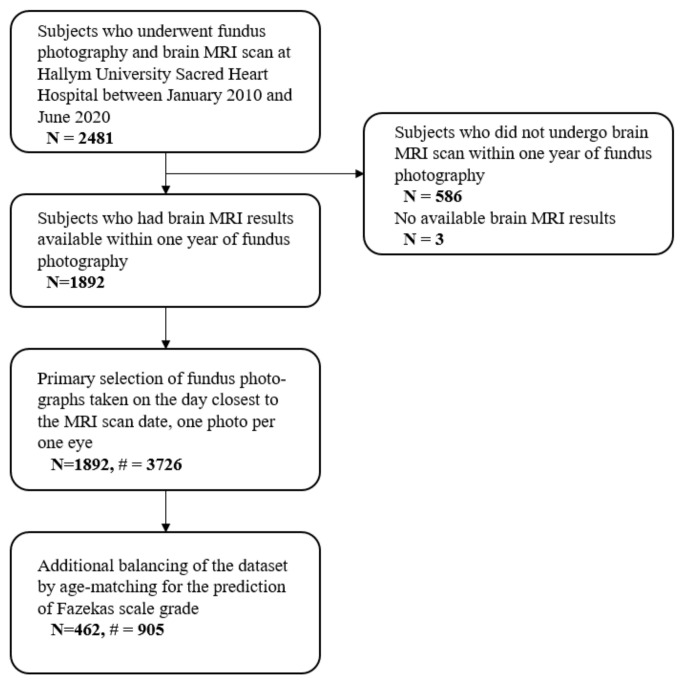
Flowchart of participant enrollment. #, number of photograph.

**Figure 2 jcm-11-03309-f002:**
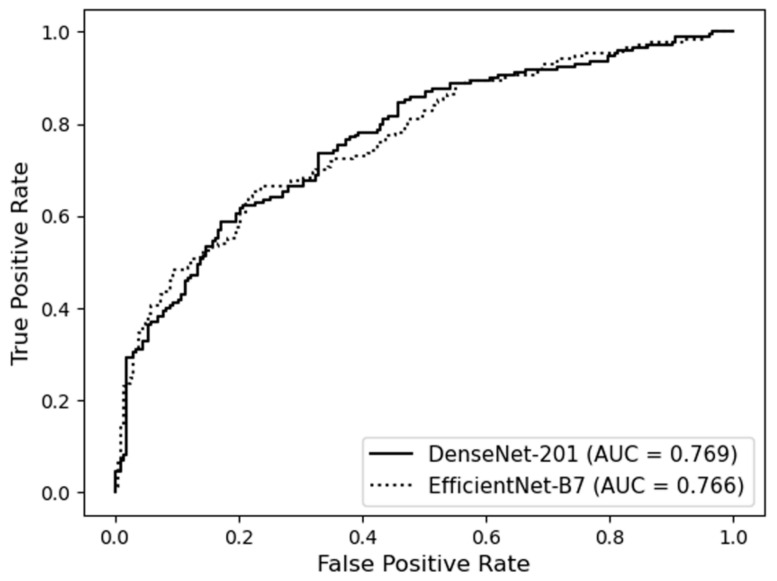
Receiver operating characteristic curves of deep learning models predicting the presence of white matter hyperintensity.

**Figure 3 jcm-11-03309-f003:**
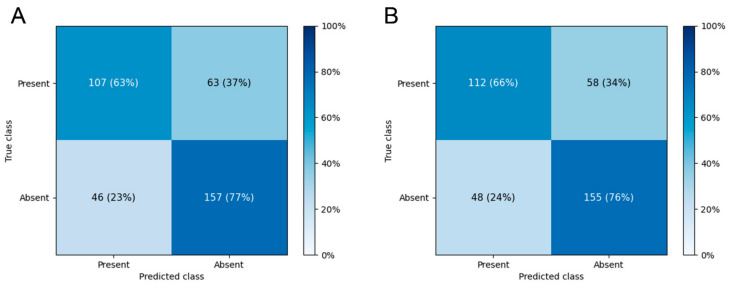
Confusion matrices of the best-performing folds of (**A**) DenseNet-201 and (**B**) EfficientNet-B7 for the prediction of the presence of white matter hyperintensity.

**Figure 4 jcm-11-03309-f004:**
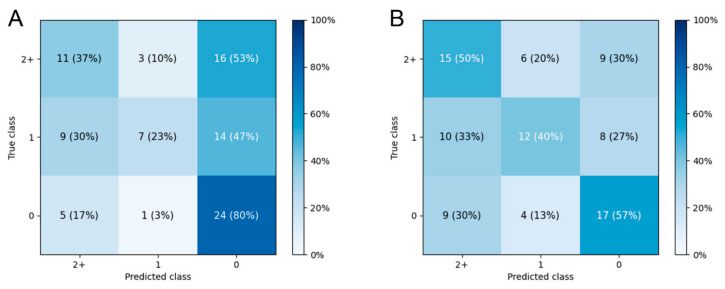
Confusion matrices of the best-performing folds on (**A**) DenseNet-201 and (**B**) EfficientNet-B7 for the FS grade prediction.

**Figure 5 jcm-11-03309-f005:**
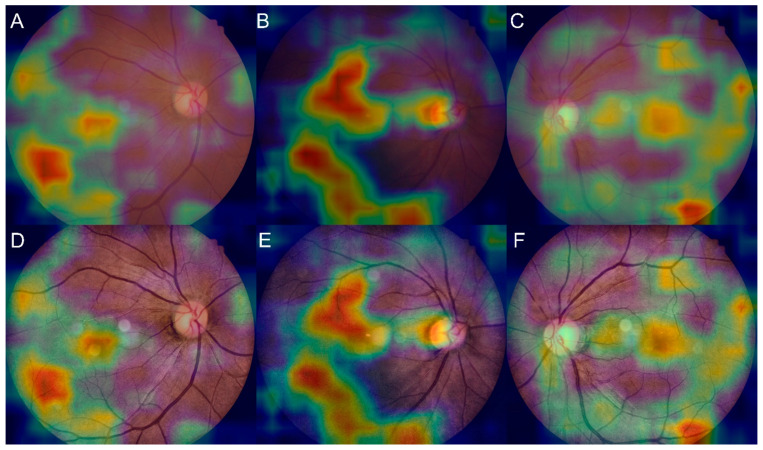
Representative examples of salience maps showing the area of strong attention as red for FS grade 2+ over (**A**–**C**) original fundus images and (**D**–**F**) pre-processed images.

**Table 1 jcm-11-03309-t001:** Composition of the fundus photograph dataset.

	Whole Dataset		Training Dataset		Test Dataset	
	Fundus N	Patients N	Fundus N	Patients N	Fundus N	Patients N
Prediction of the presence of WMH						
Overall	3726	1892	3353	1703	373	189
Absent WMH	2024	1019	1821	917	203	102
Present WMH	1702	873	1532	786	170	87
Prediction of Fazekas scale						
Overall	905	462	814	416	91	46
Grade 0	303	154	272	138	31	16
Grade 1	302	154	272	139	30	15
Grade 2+	300	154	270	139	30	15

N, number; WMH, white matter hyperintensity.

**Table 2 jcm-11-03309-t002:** Diagnostic performance of deep learning models predicting the presence of white matter hyperintensity.

Model	Diagnostic Performance, % (95% CI)				AUC (95% CI)
	Sensitivity	Specificity	PPV	NPV	
DenseNet-201	66.1 ± 8.8	71.3 ± 8.0	66.5 ± 4.1	71.8 ± 3.6	0.736 ± 0.030
EfficientNet-B7	61.7 ± 9.8	73.9 ± 7.4	66.9 ± 3.5	70.1 ± 4.2	0.724 ± 0.026

Model performances are presented as percent (%). CI, confidence interval; PPV, positive predictive value; NPV, negative predictive value; AUC, area under the curve.

**Table 3 jcm-11-03309-t003:** Diagnostic precision values predicting the presence of white matter hyperintensity in the 10-fold cross validation.

Model	Fold 0	Fold 1	Fold 2	Fold 3	Fold 4	Fold 5	Fold 6	Fold 7	Fold 8	Fold 9
DenseNet-201	0.721	0.655	0.635	0.729	0.653	0.617	0.649	0.714	0.620	0.657
EfficientNet-B7	0.674	0.628	0.692	0.714	0.689	0.626	0.692	0.710	0.626	0.643

## Data Availability

The data are available from the corresponding author on reasonable request.

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
