# Peer review of "Prediction of White Matter Hyperintensity in Brain MRI Using Fundus Photographs via Deep Learning"

_jcm, 2022, doi:10.3390/jcm11123309_

Round 1
Reviewer 1 Report
Abstract should be rewritten in a more abstract-style it seems like a part of the conclusions section.
Introduction should present what deep learning is and what benefits it has. Also a bigger state of the art should be given.
In methods section, it talks about the unbalanced dataset which latter seems to be balanced but no explanations are stated about how this has been done. Authors talk about to different classifications but they are not introduce before and not explained in-depth. Models must be tested using a training, validation, test and metrics about these stages must be given. In particular, they have to accomplish with the bias-variance trade off to avoid over and underfitting. During training stage, it must be explained if grid search strategies have been used to fine tune the hyperparamaters. Architectures and its formal definitions must be given. Confusion matrices must be interpreted in-depth in which cases are the models failing? which consequences do they have in the patients?
Finally, future works must be provided.
Author Response
The authors appreciate the editor and reviewers for their considerate comments. According to the recommendations, we have carefully reread and rewritten our manuscript. Our point-by-point responses to the reviewers’ comments are as follows.
First of all, we found an error in the process of splitting the training dataset and the tuning dataset, which was that the splitting was not performed as class-stratified. Thus, we corrected the error and re-performed the 10-fold cross-validation experiments throughout our article, as well as the 5-fold cross-validation experiments requested by the reviewers. The results were nearly as same as the prior results but slightly improved. We revised our manuscript according to the new results.
Reviewer #1.
1) Abstract should be rewritten in a more abstract-style it seems like a part of the conclusions section.
: Thank you for the comment. As the reviewer’s suggestion, we revised the abstract as below:
Purpose: We investigated whether the deep learning of retinal fundoscopic images could predict cerebral white matter hyperintensity (WMH), as represented by the modified Fazekas scale (FS) in brain magnetic resonance imaging (MRI). Methods: Participants who had undergone brain MRI and health-screening fundus photography at Hallym University Sacred Heart Hospital between 2010 and 2020, were consecutively included. The subjects were divided by the presence of WMH, and also classified into three groups according to the FS grade (0 vs. 1 vs. 2+) using age-matching. Two pre-trained convolutional neural networks were fine-tuned and evaluated for prediction performance using 10-fold cross-validation. Results: A total of 3,726 fundus photographs from 1,892 subjects were included, of which 905 fundus photographs from 462 subjects were included in the age-matched balanced dataset. In predicting the presence of WMH, the mean area under the receiver operating characteristic curves was 0.736 ± 0.030 for DenseNet-201, and 0.724 ± 0.026 for EfficientNet-B7. For the prediction of FS grade, the mean accuracies reached 41.4 ± 5.7% by DenseNet-201 and 39.6 ± 5.6% by EfficientNet-B7. The deep learning models focused on the macula and retinal vasculature to detect an FS of 2+. Conclusions: Cerebral WMH might be partially predicted by non-invasive fundus photography via deep learning, which may suggest an eye-brain association.
2) Introduction should present what deep learning is and what benefits it has. Also a bigger state of the art should be given.
: Thank you for your insightful comments. We added sentences that elaborate on the characteristics and benefits of deep learning approach with state of the art references (Line No. 64-78).
Furthermore, with recent advances in computing power and storage devices, deep learning (DL) algorithms, an application of artificial intelligence, have been widely adopted in medical research, including in medical image evaluation, sentiment analysis using medical texts, and outcome prediction using sequential signal data. As DL algorithms may accurately incorporate multiple imaging features which are invisible to humans, previous studies have utilized several DL models to reveal the predictability of fundus photography on a variety of ocular diseases, including age-related macular degeneration, glaucoma, and central retinal vein occlusion. Recently, retinal characteristics found using machine learning models for automatic retinal image analysis (ARIA) have been found to predict WMH and its classification in healthy adults. However, no study has yet investigated whether raw fundoscopy images that use a DL approach have potential for predicting degree of cerebral WMH in the general population.
3) In methods section, it talks about the unbalanced dataset which latter seems to be balanced but no explanations are stated about how this has been done. Authors talk about to different classifications but they are not introduce before and not explained in-depth.
: Thank you for the considerate comment. We admit that we left out the detailed process of data balancing. Actually, all patients were divided into age subgroups at intervals of 5 years, which started from the 15-19 years age subgroup and ended at the 85-89 years age subgroup. Then, for each age subgroup, random sampling was performed for each FS grade class as much as the number of patients of the class with the smallest number of patients. The detailed profiles of age subgroups are presented below. Consequently, 154 patients were included in each FS grade group. We added this detailed explanation in the Method section of the manuscript (Line 157-165). We appreciate your comment.
Age, years |
Number of patients |
Minimal number of patients |
||
|
FS grade 0 |
FS grade 1 |
FS grade 2 |
in each FS group |
15-19 |
2 |
1 |
0 |
0 |
20-24 |
3 |
2 |
0 |
0 |
25-29 |
4 |
0 |
0 |
0 |
30-34 |
24 |
4 |
0 |
0 |
35-39 |
54 |
3 |
0 |
0 |
40-44 |
109 |
21 |
1 |
1 |
45-49 |
186 |
47 |
4 |
4 |
50-54 |
254 |
108 |
16 |
16 |
55-59 |
217 |
149 |
31 |
31 |
60-64 |
96 |
118 |
35 |
35 |
65-69 |
50 |
94 |
47 |
47 |
70-74 |
17 |
64 |
47 |
17 |
75-79 |
3 |
23 |
39 |
3 |
80-84 |
0 |
6 |
10 |
0 |
85-89 |
0 |
1 |
2 |
0 |
Supplemental figure for distribution of FS according to age subgroup
The main outcome measure was the predictive performance for the presence of WMH. WMH was regarded to be present only at the FS grade 1 or higher, as defined by the Fazekas scale [18]. The secondary outcome measure was the predictive performance for the grade of FS as grades 0, 1, or 2+ based on fundus photographs. Notably, there was a significant between-group data imbalance for the patient age as well as for the data size (Figure S1). Therefore, for the FS grade prediction, to verify the predictability and avoid the age bias, the FS grade 0, 1, and 2+ groups were age-matched via random under-sampling (Figure 1). For this dataset balancing, all patients were divided into age subgroups at intervals of 5 years, which started from the 15-19 years age subgroup and ended at the 85-89 years age subgroup. Then, for each age subgroup, random sampling was performed for each FS grade class as much as the number of patients of the class with the smallest number of patients.
4) Models must be tested using a training, validation, test and metrics about these stages must be given. In particular, they have to accomplish with the bias-variance trade off to avoid over and underfitting.
: Thank you for the thoughtful comment. Another reviewer raised same issues for the bias-variance trade off for this experiment. After discussing this issue with our participating authors, we concluded that it is best to present the performance of each validation fold of the 10-fold cross validations experiments. We added this result in Table 3 (Line 273)
CV fold |
0 |
1 |
2 |
3 |
4 |
5 |
6 |
7 |
8 |
9 |
DenseNet 201 |
0.721 |
0.655 |
0.635 |
0.729 |
0.653 |
0.617 |
0.649 |
0.714 |
0.620 |
0.657 |
EfficientNet b7 |
0.674 |
0.628 |
0.692 |
0.714 |
0.689 |
0.626 |
0.692 |
0.710 |
0.626 |
0.643 |
Result of precision of each cross-validation fold in DenseNet and EfficientNet classifiers.
5) During training stage, it must be explained if grid search strategies have been used to fine tune the hyperparamaters. Architectures and its formal definitions must be given.
: We appreciate your professional comment. For hyperparameter tuning, genuine grid search strategies were not applied. However, we explored the best performing condition of hyperparameters by sequential reducing. For batch size, the largest size that the GPU memory could deal with for EfficientNet-B7 in our server was 6. We explored the performance of each batch size, decreasing it to 6, 4, and 3, in our pilot studies, and then adopted the best performing batch size, 4. After fixing the batch size, the learning rate was explored starting from 1e-4 and then being sequentially reduced by 1/10. Finally, the learning rate, 1e-6, which showed the highest performance, was adopted. We added this description in the Method section of the manuscript (line 196-205).
In terms of the architectures, we used the architectures open to the public. We already described the references and brief explanations or definitions in our manuscript (line 191-194). However, we added the original source of the architectures in the Method section (line 194-196). We hope this will meet your request. Thank you for the comment. (Line 193- 206)
Cross entropy function is widely used in deep neural network classification, in which minimizing cross entropy effects well to discriminative between correct and competing classes with non-linear fashion. [25].
6) Confusion matrices must be interpreted in-depth in which cases are the models failing? which consequences do they have in the patients?
: Thank you for the comment. In additional assessment for the result of confusion matrix, we identified that there was a statistical difference of age distribution in each cell. We additionally reported this in Table S2
7) Finally, future works must be provided.
: Thank you for your thoughtful comment. In the future, the research involving with a larger number of patients NOT from health screening center BUT from out-patient clinic will be required, thus to include a larger number of patients having a high grade of Fazekas scale (2 or 3). Also, various diseases showing detectable changes in the brain MRI scan, such as Alzheimer’s disease, could be investigated for the association with retinal changes. We added this point in the Discussion according to your advice (line 362-376, 396-404).
They suggested that changes in retinal vessels were observed earlier in patients with cognitive decline related to SVD than in patients with normal cognition. That is, retinal vessel change was more closely related to actual CVD disease development than a surrogate CVD marker, such as SVD. Besides, the saliency map was also changed as a result of 2-class classification. Saliency map, macula and blood vessels were identified as a region of interest that plays an important role in WMH classification. In other words, changes in macula or blood vessels would be affect the binary classification more than pathological changes in the entire retina. Recently, it has been reported that the risk of CVD can be predicted more sensitively by DL when precise ocular images are used, such as optical coherence tomography, rather than fundus images [38]. We assume that DL prediction for SVD burden using additional ophthalmic images will improve our model’s performance in later trials.
In the future, the researches with a larger number of patients recruited from out-patient clinics would be required, to include a larger number of patients having a high grade of Fazekas scale (2 or 3). Also, various diseases showing detectable changes in the brain MRI scan, such as Alzheimer’s disease, could be investigated for the association with retinal changes using DL. There will be more reports on the relevance of fundus image DL and its prediction of other extraocular or systemic diseases in future studies. Our research methods may serve as a framework for basic analysis methods in such studies.
Reviewer 2 Report
The manuscript entitled "Prediction of White Matter Hyperintensity in Brain MRI using Fundus Photographs via Deep Learning” has been investigated in detail. The topic addressed in the manuscript is potentially interesting and the manuscript contains some practical meanings, however, there are some issues which should be addressed by the authors:
- Altough the meaning of flowchart of Fig. 1 can be understood but it could be improved.
- The data has been divided to test data and train data. According to lots of references the ratio is considered 70% for training and 30% for testing. Please describe the considered percentages and the reason with related reference.
- The authors should clearly emphasize the contribution of the study. Please note that the up-to-date of references will contribute to the up-to-date of your manuscript. The Artificial Intelligence based studies in 2021 such as “Application of GMDH neural network technique to improve measuring precision of a simplified photon attenuation based two-phase flowmeter” could be used in the study or to indicate the contribution in the "Introduction" section.
- The authors states that K-fold cross validation method was used in the paper. Please add the related result in a separate figure for this method which shows the precision in every k cases.
- Would you please suggest several methods for improving the results in conclusion section? The idea and the presented work is really interesting but the results could be improved surely.
This study may be consider for publication if it is addressed in the specified problems.
Author Response
The authors appreciate the editor and reviewers for their considerate comments. According to the recommendations, we have carefully reread and rewritten our manuscript. Our point-by-point responses to the reviewers’ comments are as follows.
First of all, we found an error in the process of splitting the training dataset and the tuning dataset, which was that the splitting was not performed as class-stratified. Thus, we corrected the error and re-performed the 10-fold cross-validation experiments throughout our article, as well as the 5-fold cross-validation experiments requested by the reviewers. The results were nearly as same as the prior results but slightly improved. We revised our manuscript according to the new results.
Reviewer #2.
The manuscript entitled "Prediction of White Matter Hyperintensity in Brain MRI using Fundus Photographs via Deep Learning” has been investigated in detail. The topic addressed in the manuscript is potentially interesting and the manuscript contains some practical meanings, however, there are some issues which should be addressed by the authors:
1) Although the meaning of flowchart of Fig. 1 can be understood but it could be improved.
: We appreciate your kind comments improving our manuscript. According to your advice, we revised Figure 1, in a more understandable form. We hope this will meet your expectation. If there is anything to be more changed, please let us know. Thank you.
2) The data has been divided to test data and train data. According to lots of references the ratio is considered 70% for training and 30% for testing. Please describe the considered percentages and the reason with related reference.
: Thank you for your pertinent comment. We also agreed that most of training dataset of deep learning for image classification is suitable for 70~90% of the whole dataset. Ten percent of the training-testing splitting is also used in a variety of image classification task as below references.
â‘ Lee K, Lee K, Lee H, Shin J. A simple unified framework for detecting out-of-distribution samples and adversarial attacks. Advances in neural information processing systems. 2018;31.
② Pham TD. Classification of COVID-19 chest X-rays with deep learning: new models or fine tuning?. Health Information Science and Systems. 2021 Dec;9(1):1-1.
③ Killamsetty K, Durga S, Ramakrishnan G, De A, Iyer R. Grad-match: Gradient matching based data subset selection for efficient deep model training. In International Conference on Machine Learning 2021 Jul 1 (pp. 5464-5474). PMLR.
④ Medina G, Buckless CG, Thomasson E, Oh LS, Torriani M. Deep learning method for segmentation of rotator cuff muscles on MR images. Skeletal Radiology. 2021 Apr;50(4):683-92.
We tried to add these references additionally into the revised manuscript. However, as the number of allowed references increased, we did not cite additionally.
3) The authors should clearly emphasize the contribution of the study. Please note that the up-to-date of references will contribute to the up-to-date of your manuscript. The Artificial Intelligence based studies in 2021 such as “Application of GMDH neural network technique to improve measuring precision of a simplified photon attenuation based two-phase flowmeter” could be used in the study or to indicate the contribution in the "Introduction" section.
: Thank you for your insightful comments. As advised, we have emphasized the potential contribution of our study by adding the state of art references including one the reviewer has recommended in the “Introduction” section (Line 70).
As DL algorithms may accurately incorporate multiple imaging features which are invisible to humans [12,13], previous studies have utilized several DL models to reveal the predictability of fundus photography on a variety of ocular diseases, including age-related macular degeneration, glaucoma, and central retinal vein occlusion [14,15].
Reference No. 13
4) The authors states that K-fold cross validation method was used in the paper. Please add the related result in a separate figure for this method which shows the precision in every k cases.
: Thank you for your considerate comment. According to your advice, we added the Table 3, which shows the precision in every k cases, in the manuscript (line 273).
Table 3. Diagnostic precision values predicting the presence of white matter hyperintensity in the 10-fold cross validation.
Model |
Fold 0 |
Fold 1 |
Fold 2 |
Fold 3 |
Fold 4 |
Fold 5 |
Fold 6 |
Fold 7 |
Fold 8 |
Fold 9 |
DenseNet-201 |
0.721 |
0.655 |
0.635 |
0.729 |
0.653 |
0.617 |
0.649 |
0.714 |
0.620 |
0.657 |
EfficientNet-B7 |
0.674 |
0.628 |
0.692 |
0.714 |
0.689 |
0.626 |
0.692 |
0.710 |
0.626 |
0.643 |
5) Would you please suggest several methods for improving the results in conclusion section? The idea and the presented work is really interesting but the results could be improved surely. This study may be consider for publication if it is addressed in the specified problems.
: Thank you for your suggestion. We thought that the weakness of this paper was related to the low performance of 3-class classification commonly pointed out by all of the reviewers. Therefore, we have revised the revised manuscript considerably to explain the results of 2-class classification. Please let us know if you have any additional suggestions.
Reviewer 3 Report
In the proposed research, raw fundoscopic images are used to predict cerebral SVD using DenseNet and EfficientNet deep learning approaches. The hypothesis seems valid and supports the notion but overall accuracy is far below the acceptance level. it is suggested that instead of three class division, data should be divided into two classes. Also 10 fold cross validation can be reduced to 5 fold cross validation that may improve the accuracy. Figure 1 can also be improved if fundoscopic images for each block should be displayed (one or two) along with the box.
What is the significance of time between fundoscopic image and MRI and if it is exceeded by more than a year, how it affects the results.
How the pathological retinal structures can be identified using pre-trained deep learning models.
Is there any mathematical formulation to relate input data (fundoscopic images or their features with the predicted output SVD labels. In case of non linear function, how control parameters assist in achieving the desired level of classification.
Author Response
The authors appreciate the editor and reviewers for their considerate comments. According to the recommendations, we have carefully reread and rewritten our manuscript. Our point-by-point responses to the reviewers’ comments are as follows.
First of all, we found an error in the process of splitting the training dataset and the tuning dataset, which was that the splitting was not performed as class-stratified. Thus, we corrected the error and re-performed the 10-fold cross-validation experiments throughout our article, as well as the 5-fold cross-validation experiments requested by the reviewers. The results were nearly as same as the prior results but slightly improved. We revised our manuscript according to the new results.
Reviewer #3.
1) In the proposed research, raw fundoscopic images are used to predict cerebral SVD using DenseNet and EfficientNet deep learning approaches. The hypothesis seems valid and supports the notion but overall accuracy is far below the acceptance level. it is suggested that instead of three class division, data should be divided into two classes. Also 10 fold cross validation can be reduced to 5 fold cross validation that may improve the accuracy.
: We appreciate your thoughtful comments. According to your comment, we re-tried all experiments using the 5-fold cross validation and obtained the following results.
Table S3. Prediction of the presence of WMH (binary classification) using 10-fold cross validation
DenseNet 201 |
EfficientNet B7 |
|||
|
AUC |
Accuracy |
AUC |
Accuracy |
Fold 0 |
0.736 |
0.676 |
0.712 |
0.646 |
Fold 1 |
0.725 |
0.678 |
0.690 |
0.651 |
Fold 2 |
0.750 |
0.672 |
0.726 |
0.653 |
Fold 3 |
0.750 |
0.694 |
0.718 |
0.665 |
Fold 4 |
0.763 |
0.685 |
0.748 |
0.691 |
Fold 5 |
0.734 |
0.656 |
0.723 |
0.661 |
Fold 6 |
0.683 |
0.661 |
0.692 |
0.683 |
Fold 7 |
0.769 |
0.708 |
0.766 |
0.716 |
Fold 8 |
0.687 |
0.635 |
0.712 |
0.665 |
Fold 9 |
0.761 |
0.689 |
0.756 |
0.708 |
Performance average ± SD (95% CI) |
0.736±0.030 (0.733~0.739) |
0.676±0.021 (0.674~0.678) |
0.724±0.026 (0.721~0.727) |
0.674±0.024 (0.672~0.676) |
Table S4. Prediction of the presence of WMH (binary classification) using 5-fold cross validation
DenseNet 201 |
EfficientNet B7 |
|||
|
AUC |
Accuracy |
AUC |
Accuracy |
Fold 0 |
0.720 |
0.658 |
0.701 |
0.638 |
Fold 1 |
0.746 |
0.670 |
0.729 |
0.659 |
Fold 2 |
0.753 |
0.693 |
0.734 |
0.691 |
Fold 3 |
0.735 |
0.687 |
0.714 |
0.679 |
Fold 4 |
0.715 |
0.651 |
0.688 |
0.646 |
Performance average ± SD (95% CI) |
0.734 ± 0.016 (0.733~0.735) |
0.672 ± 0.018 (0.671~0.673) |
0.713 ± 0.019 (0.712~0.714) |
0.663 ± 0.022 (0.661~0.665) |
Table S5. Prediction of the Fazekas scale grade (3-class classification) using 10-fold cross validation
|
DenseNet 201 |
EfficientNet B7 |
|
Accuracy |
|
Fold 0 |
0.451 |
0.385 |
Fold 1 |
0.440 |
0.429 |
Fold 2 |
0.411 |
0.389 |
Fold 3 |
0.367 |
0.422 |
Fold 4 |
0.451 |
0.396 |
Fold 5 |
0.286 |
0.352 |
Fold 6 |
0.378 |
0.456 |
Fold 7 |
0.467 |
0.489 |
Fold 8 |
0.467 |
0.333 |
Fold 9 |
0.429 |
0.308 |
Performance average ± SD (95% CI) |
0.414±0.057 0.402 ~0.426 |
0.396 ± 0.056 0.385 ~0.407 |
Table S6. Prediction of the Fazekas scale grade (3-class classification) using 5-fold cross validation
|
DenseNet 201 |
EfficientNet B7 |
|
Accuracy |
|
Fold 0 |
0.459 |
0.414 |
Fold 1 |
0.385 |
0.396 |
Fold 2 |
0.354 |
0.392 |
Fold 3 |
0.428 |
0.367 |
Fold 4 |
0.381 |
0.409 |
Performance average ± SD (95% CI) |
0.401±0.042 0.395 ~0.407 |
0.396±0.018 0.393~0.399 |
To summarize, the 5-fold cross validation could not increase the prediction performance in comparison with 10-fold cross validation. Indeed, the 5-fold cross validation slightly decreased the performances of deep learning models. We would think that this might be because the 5-fold cross validation method used a less number of data in the training dataset in each fold. Anyway, the 5-fold cross validation did not work well, so we did not add the results in the manuscript. We hope you can kindly understand this issue. We appreciate your insightful advice.
2) Figure 1 can also be improved if fundoscopic images for each block should be displayed (one or two) along with the box.
: Thank you for the helpful comment. We carefully remade the Figure 1 to show the inclusion process more clearly. Nevertheless, there were no distinct differences in the funduscopic images between each step. Thus, we could not display the appropriate images. Instead, we changed the Figure 1 in a more easily understandable form. If there is any good recommendation, please let us know. Thank you.
3) What is the significance of time between fundoscopic image and MRI and if it is exceeded by more than a year, how it affects the results.
: Thank you for the comment. Performance of DL algorithm would be underestimated as the time difference between MRI and fundoscopic exam increased. Therefore, we selected the MRI and fundoscopic datasets with the closest time difference between MRI and fundoscopic examination within 1 year. Therefore, we added this reference in the method section as below (Line 104)
Since this study was not to assess retinopathic change in patients with confirmed SVD on MRI, there is a time difference between the MRI and fundoscopic examination. As the time of fundoscopy is later than the time of MRI, fundoscopic changes may be overestimated in the ML model using MRI dataset. Therefore, we only included the subjects with a difference between MRI and fundus imaging within 1 year.
4) How the pathological retinal structures can be identified using pre-trained deep learning models.
: Thank you for the comment. In our revised manuscript, the saliency map was also changed as a result of 2-class classification. In the case of patients with WMH, macula and blood vessels were identified as a region of interest that plays an important role in WMH classification. In other words, changes in macula or blood vessels would be affect the binary classification more than pathological changes in the entire retina.
So, we added this sentence in discussion section as below: (Line 362)
Our CNN algorithms showed a slightly lower sensitivity for predicting high FS in two- and three-class classification. In other words, the predicted modified FS values were lower than the actual modified FS values. The performance of our algorithm could be improved, but the current results have the following possibilities if we consider the results of previous studies on SVD. Previous pathology studies on WMH have revealed that WMH development is not solely linked to demyelination and axonal loss due to ischemic change. Subtler WMH was also associated with endothelial and microglia activation, implicating the association of WMH with neuroinflammation. These complex pathomechanisms of WMH on brain MRI may have reduced the power and sensitivity of our DL models using Fundoscopy. In further studies on SVD and retinopathy, retinal vessel changes are evident only after overt SVD development, such as cognitive impairment. Nunley et al. investigated 119 patients over c18 years of age to analyze the long-term relationship between retinal vessel changes and cognitive impairment in patients with Type 1 diabetes [31]. They suggested that changes in retinal vessels were observed earlier in patients with cognitive decline related to SVD than in patients with normal cognition. That is, retinal vessel change was more closely related to actual CVD disease development than a surrogate CVD marker, such as SVD. Besides, the saliency map was also changed as a result of 2-class classification. Saliency map, macula and blood vessels were identified as a region of interest that plays an important role in WMH classification. In other words, changes in macula or blood vessels would be affect the binary classification more than pathological changes in the entire retina. Recently, it has been reported that the risk of CVD can be predicted more sensitively by DL when precise ocular images are used, such as optical coherence tomography, rather than fundus images [32]. We assume that DL prediction for SVD burden using additional ophthalmic images will improve our model’s performance in later trials.
5) Is there any mathematical formulation to relate input data (fundoscopic images or their features with the predicted output SVD labels. In case of non-linear function, how control parameters assist in achieving the desired level of classification.
: Thank you for the professional comment. In this study, we selected a cross entropy function as a loss function to optimize our model. From a probabilistic point of view, if we denote the input data (fundoscopic images) as ‘X’, SVD labels as ‘Y’ and parameters as ‘θ’ then, likelihood can represents P(Y|X;θ) and we would like to find θ to maximize likelihood. As first mentioned, the cross entropy function is used as a method to calculate the difference between two probability distributions ‘p’ and ‘q’ based on information theory.
Cross entropy = -Σx p(x) log q(x)
Consider p(x) as p(y|x) and q(x) as p(y|x;θ), and we can rewrite cross entropy like -Σx p(y|x) log p(y|x;θ). It is the same as expectation of negative log likelihood and from the point of view of maximizing likelihood equals the expectation of log likelihood with the argument θ. Therefore, it is equivalent to finding theta that minimizes cross entropy and negative log likelihood.
In this way, the parameters were tuned using a mathematical formulation that satisfies both the probabilistic and information theory perspectives. We briefly added this explanation in the manuscript (Line 206). Thank you.
Cross entropy function is widely used in deep neural network classification, in which minimizing cross entropy effects well to discriminative between correct and competing classes with non-linear fashion. [25].
Round 2
Reviewer 1 Report
Please add this paper
LeCun, Y., Bengio, Y., & Hinton, G. (2015). Deep learning. nature, 521(7553), 436-444.
Reviewer 2 Report
My recommendation is "Accept".